# OpenReview forum: "Language Model Inversion"
_ICLR.cc/2024/Conference — ICLR 2024 poster_

### Official Review · Reviewer_scV6 · 2023-10-29

**Soundness:** 3 good
**Presentation:** 2 fair
**Contribution:** 2 fair
**Rating:** 5
**Confidence:** 4

**Summary:**

The paper trains an encoder-decoder model to map an embedding back to the originating text. The problem and the method are very similar to [1]. The main differences seem to be

(1) The paper is motivated in the context of LLMs (extract underlying prompts), whereas [1] is in the context of retrievers (extract paragraphs in the knowledge base).

(2) In [1], the embedding is a low-dimensional text embedding of a retriever. Here, the embedding is the conditional token logits given the prompt, which are unrolled and transformed to be treated as encoder inputs.

(3) This assumes access to LLM probabilities, which may not be entirely available. The paper proposes an estimator based on bisection that can be done with API calls, but this also assumes at least top-1 logit is exposed.

(4) A critical component in [1] is the iterative refinement, which is not considered in this paper.

[1] Text Embeddings Reveal (Almost) As Much As Text (to appear at EMNLP)

**Strengths:**

- Model inversion is a relatively new interesting problem.
- The proposed method is shown to be more effective than more naive versions (specifically, not using the token logits/Monte Carlo estimator for probabilities).

**Weaknesses:**

- Given the high similarity between [1] and this submission, every nook and cranny of their differences must be disclosed and analyzed to make the submission's contributions clear. I found comparisons lacking with many unanswered questions like: why not use the iterative version, which is shown to be extremely useful in [1]?
- The bisection-based estimator seems effective, but I find it a bit abrupt and lacking an explanation.
- The method does not seem to consider the setting in which the model probabilities are not available at all (i.e., generation-only APIs), which is maybe the most relevant setting for extracting prompts given that the most powerful LLMs are private.

**Questions:**

See above.

---

> ### Author Response · Authors · 2023-11-22
> **Official Response to Reviewer scV6**
>
> >  The paper proposes an estimator based on bisection that can be done with API calls, but this also assumes at least top-1 logit is exposed.
>
> To clarify, we do not assume the top-1 logit is exposed, only the argmax output (which is given in all APIs we know of). Our only assumption is that we can provide a "logit bias" argument. We do not require the API to return logits or log probabilities.
>
> > Given the high similarity between [1] and this submission, every nook and cranny of their differences must be disclosed and analyzed to make the submission's contributions clear. I found comparisons lacking with many unanswered questions like: why not use the iterative version, which is shown to be extremely useful in [1]?
>
> We agree that the method here is not novel. We apply ideas from past inversion work including [1] to a different application domain (inverting LM outputs). The contributions are developing a new architecture, a dataset of LM prompts, new baselines, and an algorithm for making this practical to run in an API setting with logit-bias.
>
> We tested the iterative refinement approach in [1] but unfortunately did not find it worked when the input is LM logits. We have added these negative results to the appendix, along with a small experiment explaining why our corrector module might not work recursively (due to a narrow range of BLEU scores in the training distribution).
>
> > The bisection-based estimator seems effective, but I find it a bit abrupt and lacking an explanation.
>
> We’ve completely rewritten Section 5 in an effort to make our logit recovery algorithm more clear, and smooth the transition between sections. We plan to release our algorithm as a Python package to allow others to recover full probability vectors from argmax-only API access.
>
> > The method does not seem to consider the setting in which the model probabilities are not available at all (i.e., generation-only APIs), which is maybe the most relevant setting for extracting prompts given that the most powerful LLMs are private.
>
> Sorry for the confusion. We *do* consider this baseline – “Sample Inverter” is a T5 encoder-decoder trained to directly predict a prompt from LLAMA-2 Chat outputs. We have edited the paper to make this more clear in both the theoretical discussion (Section 3.2) and presentation of results (Section 7).
>
> Additionally, the algorithm in Section 5 (originally known as “Distribution Extraction”) is proposed for precisely this reason: it enables logits access to models when APIs only provide argmax directly, by use of the logit bias argument. (As of November 2023, logit bias and argmax output are available in OpenAI, Anthropic, Cohere APIs.)

---

### Official Review · Reviewer_YYeq · 2023-10-31

**Soundness:** 3 good
**Presentation:** 2 fair
**Contribution:** 3 good
**Rating:** 6
**Confidence:** 4

**Summary:**

This paper explores the concept of inversion from language model outputs. The main goal is to retrieve the hidden prompts from language model systems, even without direct access to model output distributions. The authors have presented various techniques and experiments to substantiate their findings and have approached the problem from both attack and defense perspectives.

**Strengths:**

Originality: The paper delves into a novel area, focusing on inverting language model outputs, which is not extensively studied before. This introduces a new perspective in understanding language models and their vulnerabilities.

Quality: The experiments are comprehensive and cover a range of scenarios, making the results more robust.

Significance: The findings could have implications for the safety and reliability of using large language models in real-world applications.

Clarity: The paper is generally well-structured with clear sections detailing the problem, methodology, experiments, and conclusions.

**Weaknesses:**

The descriptions of the algorithm and the results, e.g. Algorithm 1 and Figure 2, are not very clear. It's hard to fully grasp the methodology and the results without a more detailed explanation.

While the paper introduces a novel concept, the execution in terms of explaining the methodology and results could be enhanced. The paper would benefit from more detailed and clearer descriptions.

**Questions:**

Can the authors provide a more detailed explanation of Algorithm 1 and its workings? How were the models chosen for the experiments, and can the authors provide a more in-depth description of these models?

How do the results in Figure 2 correlate with the methodology described? It would be helpful to have a clearer description or perhaps an example to illustrate the findings.

---

> ### Author Response · Authors · 2023-11-22
> **Official Response to Reviewer YYeq**
>
> > The descriptions of the algorithm and the results... are not very clear. It's hard to fully grasp the methodology and the results without a more detailed explanation.
>
> Thanks for the suggestion. We have renamed the algorithm to “API-Based Logits Extraction”, but are open to suggestions for clearer terminology here. We rewrote the pseudocode and the explanatory text in Section 5.
>
> > While the paper introduces a novel concept, the execution in terms of explaining the methodology and results could be enhanced. The paper would benefit from more detailed and clearer descriptions.
>
> Thank you for this recommendation. We’ve completely written Section 5 (the section about stealing logits via API) and changed our description of the method, as well as added more details interspersed throughout the methods and results sections.
>
> > Can the authors provide a more detailed explanation of Algorithm 1 and its workings?
>
> We have completely rewritten Section 5 and the pseudocode shown in Algorithm 1. Please indicate any further points of confusion so that we can continue to improve our explanations.
>
> > How were the models chosen for the experiments, and can the authors provide a more in-depth description of these models?
>
> We chose the 7B parameter versions of LLAMA-2, and consider both the chat and base checkpoints. We chose LLAMA-2 as it was the among most powerful base language models available at the time of submission. We have clarified this point in experimental design.
>
> For the ablations in the appendix (previously Section 9), we use GPT-2 instead of LLAMA due to computational constraints; these models require ~20x less compute to run.
>
> > How do the results in Figure 2 correlate with the methodology described? It would be helpful to have a clearer description or perhaps an example to illustrate the findings.
>
> We rewrote the explanations and captions to better explain Figure 2. In particular, we explain these sampling-as-a-defense experiments differently in Section 8; please let us know if anything is still unclear.

---

> > ### Comment · Reviewer_YYeq · 2023-11-22
> >
> > I am pleased to see the improvements made in response to the comments on clarity and methodology. I believe these changes will greatly benefit the understanding and applicability of your research. It looks much much better now. However, regarding the algorithm's name change to “API-Based Logits Extraction”, as it wasn't a point raised in my review, I cannot comment on its suitability or effectiveness.
> >
> > I hope this clarification helps, and I look forward to reviewing the updated manuscript with the implemented changes.

---

> ### Author Response · Authors · 2023-11-23
>
> Thank you for the clarification and continued help. Hopefully the new name makes the purpose of our algorithm clearer to readers nonetheless... Please let us know if there are any improvements you recommend after reading the updated manuscript!

---

### Official Review · Reviewer_Cmp4 · 2023-11-03

**Soundness:** 2 fair
**Presentation:** 3 good
**Contribution:** 3 good
**Rating:** 5
**Confidence:** 3

**Summary:**

This work tries to reverse engineer the prompt given to a language model solely using its outputs. In the case of access to the distribution over next tokens (given a particular prefix), they transform the logits of this distribution to a sequence of embeddings, feed these embeddings to a pretrained language model, and use the model’s decoder in order to predict the prompt that lead to that distribution. They provide a number of empirical results that show the effectiveness of their proposed approach and explore methods for defending against prompt inversion. The further perform ablation studies to understand which components of their approach are necessary for its success

**Strengths:**

* The problem that the authors try to solve is quite interesting and relevant, given the large usage of prompt-based methods and the recent prevalence of “language models as a service.” This is the first work that I know of that tries to recover the prompt from output probabilities
* The work explores methods for defending against prompt inversion, which is likely of large interest to those that provide “language models as a service”
* Empirical results appear to be strong

**Weaknesses:**

* The method seems rather ad-hoc and not theoretically motivated. In the case of access to the conditional probability distribution, the method boils down to feeding (a transformed version of) the distribution back to a language model.
* The formal methodology is difficult to understand. For example, it’s unclear how the prompt is actually decoded from the embedded output probability distribution, i.e., what happens after their proposed encoding; I don’t understand what is going on in section 5. What does it mean to “control one logit?”

**Questions:**

* In the beginning of section 4 where it’s stated that you “train on samples from the conditional model”, what is “the conditional model”?
* Given that the vector v was projected from R^d to R^v (as part of the final linear layer of the generation model), the argument that projecting it back to R^d would lead to a loss of information feels a bit strange
* Where is the theoretical discussion/experiments corresponding to inversion when only the text output is available? The methodology discussed in section 5 still assumes access to probabilities from the model, which are often not available in “models as a service” platforms
* In section 4, it mentions that there is a “fixed-length input sequence of 42 words.” Is this in reference to the sequence of _embeddings_ that are fed to the model? Or is this alluding to how the embeddings are decoded into the expected input?
* How does the ability to reconstruct text change as a function of the pretrained model used to construct the distribution p(x | v)?

---

> ### Author Response · Authors · 2023-11-22
> **Official Response to Reviewer Cmp4**
>
> > Method is ad-hoc and not theoretically motivated.
>
> We have added a new Subsection 3.1 (“Logits Contain Residual Information”) providing empirical and theoretical motivation why language model logits may reveal information about past tokens.
>
> In terms of the method, we feel we are following standard practice in modern deep learning using Transformers. The main architectural challenge is how to feed in a high-dimensional vector (32,000 numbers) into a transformer. Main contributions were collecting a large dataset of prompts, extracting next-token probability distributions, and demonstrating that inversion is possible and effective in a variety of settings. We are not sure what would make a deep learning approach more theoretically motivated.
>
> > The formal methodology is difficult to understand. For example, it’s unclear how the prompt is actually decoded from the embedded output probability distribution, i.e., what happens after their proposed encoding.
>
> We havve tried to clarify this subsection of the paper; please let us know what you still find confusing. The output vector is provided to the encoder of an encoder-decoder language model, where the decoder autoregressively predicts the prompt. This approach is roughly standard in most conditional NLP systems.
>
> > I don’t understand what is going on in section 5. What does it mean to “control one logit?”
>
> We consider the most common setting of LLM APIs, where we can provide a “logit bias” argument that upweights or downweights the relative importance of a token in the output. This feature is key to our algorithm. We have completely rewritten Section 5 and Algorithm 1 to be clearer and more contiguous with the rest of the paper, which may alleviate your concern.
>
> > In the beginning of section 4 where it’s stated that you “train on samples from the conditional model”, what is “the conditional model”?
>
> This line was simply meant to indicate that we train the inversion model on output from a (conditional) language model. It was not a critical point, and we’ve removed the statement entirely.
>
> > Given that the vector v was projected from R^d to R^v (as part of the final linear layer of the generation model), the argument that projecting it back to R^d would lead to a loss of information feels a bit strange
>
> This is a small notational inconsistency. The original dimensionality d is that of the victim language model (4096 for LLAMA) while the final dimensionality is the input embedding of the inverter language model (typically 768 for T5). So in this case, projecting directly from LLAMA to T5 space would lead to a loss of dimensionality with a factor of 4096/768≈5.3.
>
> > Where is the theoretical discussion/experiments corresponding to inversion when only the text output is available? The methodology discussed in section 5 still assumes access to probabilities from the model, which are often not available in “models as a service” platforms.
>
> We did consider this baseline, sorry for not making it more clear. For Llama2-chat, we train a model to predict the prompt input given language model output samples . This is shown in Table 1 as “Sample Inverter”. We add an additional paragraph of theoretical discussion in this matter to Section 3.2 (“Inverting from outputs”).
>
> > In section 4, it mentions that there is a “fixed-length input sequence of 42 words.” Is this in reference to the sequence of embeddings that are fed to the model? Or is this alluding to how the embeddings are decoded into the expected input?
>
> This is the sequence of embeddings that is fed into the encoder of the inversion model.
>
> > How does the ability to reconstruct text change as a function of the pretrained model used to construct the distribution p(x | v)?
>
> We consider this question through many ablations shown in Table 9. Notably, we conclude that we could have seen greatly improved inversion performance if we had trained an inverter with more parameters.

---

### Official Review · Reviewer_xaWT · 2023-11-04

**Soundness:** 3 good
**Presentation:** 3 good
**Contribution:** 3 good
**Rating:** 6
**Confidence:** 4

**Summary:**

This paper focuses on extracting the prompt used in LM generation given access to the probability distribution of the token following the prompt. An “inverter” model is trained on pairs of distribution of distributions and prompts for this purpose. The dataset used for training is a meta dataset comprising a collections of various prompt/instruction datasets. A distribution extraction algorithm is also experimented with in case the LM API only provides access to top-k tokens and their logprobs instead of a full distribution. This approach is compared to several reasonable baselines on the task of inverting prompts of a LLaMA-2 (2B) model and training a much smaller model like T-5. Ablation analysis and other analysis around variation with inverter size, out-of-domain prompts etc. is also reported.

**Strengths:**

– The paper tackles an interesting problem with a sound approach.

– The paper is well-organized and the presentation is clear.

– The baselines are reasonable and well-designed and the approach in general outperforms these baselines.

– The quantitative analysis is sound and substantive. Ablation studies and analysis on the impact of model size, out of domain prompts, distribution extraction, using partial distribution etc. provide great insight into the results. For example, the finding that whole distribution is important for good performance and losing even the tail of distribution causes a drop in performance is very interesting and provides insights into behavior of LMs in general. I am surprised however that using unnormalized logits over logprobs causes a drop in performance as well because unnormalized logits contain more information than post softmax logprobs.

**Weaknesses:**

– While the results show a significantly better ability to recover prompts than the baseline, the absolute numbers are fairly low – the exact match scores are discouraging and the BLEU scores are also not very high. So I am not convinced if this is a serious concern for security in terms of prompt leakage. For example, looking at the qualitative examples, the inverter struggles with proper names which are often the important target tokens where security vulnerability is concerned.

– Similar to above, the performance is significantly much worse on out-of-domain prompt datasets. This indicates an inclination of the inverter model to yield a large number of false positives. I am not entirely convinced whether this is a reliable attack on LM services.

– Following the two points above, a more pointed analysis on the recoverability of sensitive information instead of recovering general purpose prompts might help establish the significance of the proposed attack more clearly. For example, can the proposed approach differentiate between or recover mildly different prompts that result in similar responses? Conversely, how good is the model at identifying seemingly benign prompt injection attacks? How good is the attack at exploiting specific sensitive information? Overall, I think the attack would be more serious if the exact match numbers are higher or exact match over sensitive tokens (like named entities) is high. Recovering paraphrased general purpose prompts while certainly interesting in its own right, doesn’t seem like a convincing threat.

– I am not entirely sure about this but the presentation gives me an impression that it only considers “first token” distribution after the prompt. This seems limited – would using token distributions at multiple positions improve the attack?

– The distribution extraction from access to argmax/top-k logits is very expensive, requiring the number of API calls equal to the size of vocabulary at least. This is likely a concern if multiple positions other than the first token are used.

**Questions:**

Please address points in the review above.

---

> ### Author Response · Authors · 2023-11-22
> **Official Response to Reviewer xaWT**
>
> Thanks for the comments and suggestions.
>
> > the absolute numbers are fairly low – the exact match scores are discouraging and the BLEU scores are also not very high. So I am not convinced if this is a serious concern for security in terms of prompt leakage.
>
> We view this work as a proof-of concept, to show that prompts do indeed leak through LM outputs, which had not been known before. Bigger models and more training would likely improve absolute numbers. The work, as is, improves upon the best jailbreaking prompts in most settings.
>
> In addition, we include new experiments that significantly improve both exact match and BLEU score for all out-of-distribution
> settings. Please see the general response for more detail.
>
> > A more pointed analysis on the recoverability of sensitive information instead of recovering general purpose prompts might help establish the significance of the proposed attack more clearly.
>
> To answer this question, we created a new dataset of prompts that include private data such as names, birthdays, and nationalities. Our models do a good job at reconstructing categorical national information (80% accuracy at countries and 50% on nationalities) while they struggle with exact numbers such as dates. Clearer analysis and examples are available in the new appendix section "Private Prompt Dataset" and the new Table 11.
>
> > Would using token distributions at multiple positions improve the attack?
>
> Inverting a prompt using probability outputs from multiple positions is an interesting suggestion. Since this setting would allow us to model the prompt with strictly more information, it would likely improve inversion performance. Although this is easy to test with local models, our threat model is the one represented by standard APIs: a single API call provides probabilities for the next token at a single position.
>
> > Distribution extraction from access to argmax/top-k logits is very expensive.
>
> We agree it has some cost and requires lots of calls, but not that it is "very expensive". Extracting the full logit distribution with argmax-only access requires a number of API calls proportional to the vocabulary size V.  For example, in the case of GPT-3.5 Instruct Turbo, using November 2023 pricing, extracting a single prompt can cost as low as $7 (`16 tokens * 3 calls per token * 100k tokens * $.0015 / 1k tokens) = $7.20`). Given the current value of these systems, this cost is not nothing, but it is also not that expensive compared to the engineering cost required to find the most effective prompt for some use cases.

---

### Author Response · Authors · 2023-11-22
**General Response to Reviewers**

We thank the reviewers for their time and extremely useful feedback. We have made significant changes to our paper based on your careful feedback, including new explanations and theoretical justification, and added new results, training two new LLAMA inversion models from scratch which produce much better BLEU scores on out-of-distribution prompts.

**Improved out-of-distribution results**: we made a small architectural change to our model that resulted in significantly improved out-of-distribution results, which appear in the new version of the draft. Here are the BLEU score changes:

LLAMA-2: Alpaca 10 -> 46; Anthropic 8 -> 25
LLAMA-2 Chat: Alpaca 35 -> 44 ; Anthropic 15  -> 26

Given this new change, the model transferability results have improved as well; the full results are available in Table 3.

**Sample inversion**: One common question from the reviewers was whether we considered inverting prompts directly from LM outputs. This was the “sample inversion” baseline in the paper; we have clarified this in several places and added some theoretical discussion to Section 3.

---

### Meta-Review · Area_Chair_sPCd · 2023-12-07

**Metareview:**

This paper proposes a method for extracting LLM prompts given the probability distribution of the token following the prompt, by training an inversion model on a collection of instruction datasets. The proposed model is compared to several baselines on the prompt inversion tasks with a few different models, including Llama-2 and Llama-2 Chat (7B). The authors have additionally trained a few more inversion models during the rebuttal period that has improved the prompt inversion performance (BLEU score).

The authors generally agree that the problem is interesting and novel, and are convinced about the effectiveness of the approach given the comparison against more naive baselines. A few concerns were raised regarding comparisons and the low BLEU scores. The authors have addressed these concerns with additional experiments (https://openreview.net/forum?id=t9dWHpGkPj&noteId=lohxI6R5ZB).

**Justification For Why Not Higher Score:**

- Reviewers are not completely convince by the comparisons.
- A few reviewers raised concerns about the clarity of the paper.

**Justification For Why Not Lower Score:**

- The problem being addressed is quite interesting.
- The authors have addressed the reviewers' main concerns with additional experiments during the rebuttal period.

---

### Decision · Program_Chairs · 2024-01-16

Accept (poster)